# Revolutionizing Grain and Particle Size Measurement in Metals: The Role of Sensor-Assisted Metallographic Image Analysis

**DOI:** 10.3390/s24165328

**Published:** 2024-08-17

**Authors:** Tushar Shirsat, Vinayak Bairagi, Amar Buchade, Ekkarat Boonchieng

**Affiliations:** 1Department Electronics and Telecommunication Engineering, AISSMS Institute of Information Technology, Pune 411001, India; rtushar.shirsat@gmail.com (T.S.); bairagi1@gmail.com (V.B.); 2Department of AI & DS, BRACT’s Vishwakarma Institute of Information Technology, Pune 411048, India; amar.buchade@viit.ac.in; 3Department of Computer Science, Faculty of Science, Chiang Mai University, Chiang Mai 50200, Thailand

**Keywords:** revolutionizing grain, particle size measurement, metallographic, sensor-assisted

## Abstract

Metallographic image analysis is vital in the field of metal science due to its potential to automate the sensing process for grain and particle size estimation. To ensure the good quality and reliability of metal products, analysis of the integrity of metallic components is required. In contemporary manufacturing processes, microscopic analysis is a crucial step, mainly when complex systems like gearboxes, turbines, or engines are assembled using various components from multiple suppliers. A final product’s quality, durability, and lifespan are determined via the quality analysis of properties of a material with proper tolerances. A flaw in a single component can cause the breakdown of the entire finished product. To ensure the good quality of a material, micro-structural analysis is necessary, which includes the routine measurement of inclusions. The particle and grain sizes of particulate samples are the most crucial physical characteristics of metals. Their measurement is routinely conducted across various industries, and they are frequently considered essential parameters in the creation of many products. This paper discusses the role of sensors in enhancing the accuracy and efficiency of metallographic image analysis, as well as the challenges and limitations associated with this technology. The paper also highlights the potential applications of sensor-assisted metallographic image analysis in various industries, such as aerospace, automotive, and construction. The paper concludes by identifying future research directions for this emerging field, including the development of more sophisticated algorithms for grain and particle size estimation, the integration of multiple sensors for more accurate measurements, and the exploration of new sensing modalities for metallographic image analysis.

## 1. Introduction

This metallographic analysis provides details about the factors directly affecting a material and its properties. It is essential to the study of metallic materials, as it allows for the measurement of particle size and morphology. An illumination system in an ultramicroscope, or a microscope with a resolution greater than 100×, enables us to view minute particles [1]. This study aims to enhance metal quality assurance by sensing metallographic images using ultramicroscopy. This will be accomplished by quantifying the microstructural properties of metals like cast iron, steel alloy, aluminum, and so forth. According to literature reviews, image processing is frequently used in metallographic microstructure analysis. The processing of morphological and counter-based images is carried out, and grain and particle sizes are computed. It is crucial to increase the effectiveness of research to a great extent. Research on metallic materials and metallographic analysis can provide information on the properties of materials, such as yield strength, tensile strength, and elongation, which greatly impact the properties of materials [2]. Traditional techniques for determining grain size are essentially manual processes, which makes them time-consuming and error-prone. Digital image processing and pattern recognition techniques have recently emerged as the primary tools for automatic quantitative metallographic analysis and grain size determination, owing to technological advancements in computing and image processing [3]. To obtain grain boundaries, grain sizes, and grain size distributions, among other factors, image analysis is beneficial in the microstructure analysis of metals [4]. Users who research finished steel products or raw steel require quality assurance. Most users come from international public and private institutions like the International Organization for Standardization (ISO) or the American Society for Testing and Materials (ASTM), and there are different stakeholders like defense agencies, automakers, and private individuals.

The suitability and utility of the quality management program are continuously evaluated through quality assurance. In many industries, it is common practice to calculate particle and grain sizes because they are important physical characteristics of particulate samples. This is frequently necessary when producing a wide range of products, but a flaw in one part still has the potential to make the entire product fail.

Due to its ability for direct particle size and morphology measurement, the microscope is a useful tool for characterizing particles. However, it is time-consuming, and manually determinations of particle size and shape using microscopy are arbitrary [5]. 

The development of microscopy was motivated by the need for a sophisticated method for quantifying the size and shape of a large-enough sample of particles to ensure statistical confidence in results. Image analysis systems should offer numerical distributions of precise shape parameters by giving quantitative and qualitative values for various shape descriptors. Numerous crucial macroscopic properties of metallic materials are influenced by their microstructure. Critical mechanical characteristics like tensile strength, elongation, and thermal and electrical properties are all directly correlated with the microstructure. Material production and manufacturing depend on an understanding of the relationship between microstructure and macroscopic properties, which can be quickly determined via ultramicroscopic imaging analysis [3].

It is challenging to distinguish grain boundaries in metallographic images with low contrast and defined boundaries. Numerous fresh image processing algorithms have been created to enhance grain size determination as a result of this research. There are still some issues, however, that require attention. As metallographic images of various alloys can have a wide range of characteristics, the main issue is that the current methods need to be universal [6]. The most common problem is that the degree of universality of existing methods is deficient due to the differences in the metallographic images of different alloys.

### Quality Assurance Parameters

Inclusions are solid substances found inside steel during production. Compounds found in steel during manufacturing are called inclusions. Metallic and non-metallic inclusions must be analyzed and recorded as part of quality control because they can significantly affect steel properties like hardness, formability, machinability, and corrosion resistance. Separating and evaluating inclusions and shape analysis can produce accurate results [7]. Additionally, updated modules use artificial intelligence to detect artifacts like scratches on the samples. Generally, the higher the steel quality, the fewer or milder the inclusions. The severity of the inclusions decreases with increasing standards of steel. As a result, non-metallic and metallic inclusions must be analyzed and recorded as part of quality management [8,9].

Porosity refers to the presence of pores within metals. Analysis of porosity is vital as it determines metal strength, and likely the corrosion and quality of metals.

Crack analysis in metals involves identifying the cause, nature, and extent of a crack or fracture in a metal component. It helps determine the root cause of failure, assess remaining strength, and prevent future failures by providing insights into the metal’s mechanical properties, environmental conditions, manufacturing processes, and service history.

In metal analysis, a flake is a thin, flat, and plate-like particle, which is found in various types of metal alloys. The presence and concentration of flakes in a metal product can have significant effects on its physical and mechanical properties, such as strength, ductility, and corrosion resistance. Flake estimation is a technique used to quantify the size and concentration of flakes in a metal sample, which is important for quality control, research and development, environmental monitoring, and failure analysis. Using ASTM standards, different types of steel substratum chemical, mechanical, and metallurgical properties are categorized, assessed, and defined. These steel standards help refineries, product manufacturers, metallurgical laboratories, and other steel end users to correctly process steel and its variants to ensure quality [10]. Figure 1 shows various quality parameters in metallic images [11].

## 2. Materials and Methods

This research study aims to develop a metallographic image analysis method based on ultramicroscopy for measuring the microstructural properties of metals like aluminum, cast iron, and steel alloys. For additional analysis, an optical microscope can be used to capture ultramicroscopic images of metals at different magnifications, including 100×, 200×, 400×, and 1000×. Figure 2 displays the block diagram of proposed research.

With automatic grain boundary reconstruction analysis, any common image can be processed. The image must be cleaned, the broken grain boundaries must be reconstructed automatically, and the image must be ready for additional automated analysis. This requires filtering with a variety of algorithms. The system receives the digital image and transforms it into grayscale. The degree of membership function, a curve between 0 and 1, which characterizes the fuzziness, is then calculated using fuzzy logic. This is used to determine whether or not a pixel is on the edge. To store fuzzy values in a fuzzy database, fuzzy logic is used via fuzzifiers and de-fuzzifiers. Only edges and corners are simple for fuzzy logic to recognize, but there is still some noise in the image. The output is then sharpened, and a clever operator removes the noise to create a consistent output. The grain size is then assessed automatically using the planimetric and intercept methods on this reconstructed image.

Software Tools used: Visual Studio C++ 2019, MATLAB 2019b, Python 3.7. 

### 2.1. Grain Size Analysis 

#### 2.1.1. Grain Size Estimation (As per ASTM Standard) [11,12,13] 

The following sample preparation steps are used for grain size estimation:
(a)Select steel alloys such as EN8, EN304, EN306, and EN31 to obtain grain size measurements; (b)Apply heat-treatment to these samples for a 2 hr temperature cycle at 850 Celsius and cool them via annealing, quenching, and normalization so as to introduce grain deformation;(c)Mount and polish all the samples, and polish the for multiple times at high grades;(d)Prepare scratchless samples and perform etching with a chemical process;(e)Capture a microscopic image and process it via further algorithmic stages to obtain grain sizes.


#### 2.1.2. Extracting Grain Boundaries by Fuzzy Logic [14,15] 

Edge pixel point P is significantly different from adjacent edge and non-edge pixels; this edge pixel is relatively faint, and the number of pixels belonging to the edge is lower. The fuzzy model is utilized to compare nearby pixels to determine whether the pixel is part of an edge or not. 

Pixel *p*(*x*, *y*) is the pixel from the digital image, as the central point. In the range of the pixel window, the *W*(*x*,*y*) size of *n* is selected as shown below: $$\left[\begin{array}{ccc} p\left(x-1,y-1\right) & p\left(x-1,y\right) & p(x-1,\;y+1)\\ p\left(x,y-1\right) & p\left(x,y\right) & p(x,\;y+1)\\ p\left(x+1,y-1\right) & p\left(x+1,y\right) & p(x+1,\;y+1) \end{array}\right]$$
Based on analysis and correlation between pixels within *W*(*x*, *y*), the fuzzy membership model, *µ*(*p*), is calculated, which measures the degree of membership:*µ*(*p*) = 1/*e*^[*p*(*x*,*y*) − *a*(*x*,*y*)]^ + 1(1)

A higher pixels membership degree indicates a higher probability that the pixel belongs to a grain boundary, where *p*(*x*,*y*) is the gray value of the center pixel in *W*(*x*,*y*). *µ*(*p*) will be a scalar value in the range of [0, 1], and *a*(*x*,*y*) will be the average value of all pixels.
(2)$$a(x,y)=\frac{1}{n^{2}}\sum_{k=x-(\frac{n-1}{2})}^{x+(\frac{n-1}{2})}\;\sum_{1=y-(\frac{y-1}{2})}^{y+(\frac{n-1}{2})}p(k,l)$$

The fuzzy model is used, and the reconstructed image can eliminate most of the noise factors. The single-scale fuzzy model is used to detect grain boundaries and can cause noise amplification. The size of the fuzzy model is adapted as shown below: *µ*(*p*) = 1/*e*^[*p*(*x*,*y*) − min [*a*(*x*,*y*).′*a*(*x*,*y*)]^ + 1 (3)
where *a*(*x*,*y*) and ′*a*(*x*,*y*) are the average values of the two windows; the higher the pixel membership degree, the higher the probability that the pixels belong to a grain boundary [16]. The edge information on the grain boundary is enhanced compared to that in traditional methods. After the images are processed through pre-processing using the fuzzy logic edge detection algorithm, grain boundaries are extracted. This method has good processing, effectively deals with a broad range of quality, and is superior to traditional methods for edge detection and reconstruction.

#### 2.1.3. Automatic Thresholding Logic

The histogram’s clear and deep valley should contain the threshold. The histogram has a large valley between two sharp peaks, which is unusual for a well-defined image. As a result, the valley area is where the ideal threshold value can be found. One method is to set a different threshold for each of the image’s RGB components, and then combine them using the AND operation. This is known as multiband thresholding.

The image’s entire brightness range is broken down into levels of gray that range from black to white. Typically, 256 gray steps are used for particle analysis. All image points and the corresponding pixels below the threshold are identified as being part of particles, and all image points above the threshold are classified as being part of noise. In thresholding, the gray values are calculated from the image by averaging the RGB levels.

After initial processing steps such as grayscale conversion, multiband thresholding, and filtering and binarization processes, reconstruction methods are applied for edge detection and reconstruction processes [17,18,19] with the gradient edge detector (GED), median edge detector (MED), gradient adaptive predictor (GAP), and the fuzzy model to enhance accuracy.

Fuzzy Model Steps [11,14]:

Step 1: Take digital input image.

Step 2: Convert digital input image into grayscale as fuzzy model requires binary values to detect edges.

Step 3: CDF (cumulative distribution factor).

Autothresholding and lognormal distribution fitting.

Step 4: Locate *p*(*x*, *y*) in window matrix.

Step 5: Calculate *d*(*i*).

Step 6: Evaluate *d*(*i*) in input membership function.

Step 7: Apply fuzzy set of rules to window matrix.

Step 8: Evaluate *d*(*i*) in input membership function.

Step 9: Defuzzification over system.

Step 10: *x* = width (Image).

True if *y = y + 1* & *x = 0*.

False if *x = x + 1*.

Step 11: *y* = height (image).

End if true.

Again, locate *p*(*x*, *y*) if false.

Reconstruct original image to perform measurement.

MLP multi-layered perception feature extraction.

#### 2.1.4. Design Calculations [11,14]

The magnification calculation is as follows:M = (Print width)/(Real width)(4)
ASTM grain size number: N = 2^((n−1)) (5)
where

N = no. of grains presented in the image

n = grain size no.
The average grain size diameter (AGD) is calculated as follows:AGD = (Total true Length)/(Grain intercepted) (6)
Total True length = (Length of lines)/M (7)
where M = magnificationWhen the number of grains increases, the grain size increases.Accuracy is determined as follows:Accuracy = (Corrected output)/(Total size) (8)



**Calculations of grain size [11,20]:**


Consider the given size of the image, with the width and height, MxN, being (165, 124):Magnification = Print width/Real width        = 165 mm/0.215 mm = 768
Number of grains in image = Whole grain + 0.5 (Partial grains)
Number of grains in image = 108 + 0.5 (32) = 140
True Area = Width/Magnification × Height/Magnification  = 165/768 × 124/768= 0.0351 mm^2

ASTM grain size number calculations
Number of grains in 1 inch square = 2^ (Grain number − 1)

A 1-inch square = 0.0645 mm^2 for 100× as per ASTM
Grain size number = Number of grains in 1-inch square × (0.0645 mm^2/True Area)
Grain size number = 140 × (0.0645/0.0351) = 257.26
log (257.26) = (n − 1) × log (2)
n = 9.07 … ASTM grain size number
Grain diameter average = True length Total/grains got intercepted.
True length total = Lines Length/magnification = 2 (124)/768 = 0.323 mm
Average grain diameter = 0.323/(16 + 14) = 0.01077 mm

The process is shown in Figure 3 for understanding purposes.

Grain size is estimated with intercept method Figure 3b where imaginary horizonal line are intercepted with reconstructed grains edges and used to calculate grain number, planimetric method provides grain number based on number of valid grains highlighted in green Figure 3c in specific region indicated with imaginary circle

### 2.2. Particle Size Analysis

The size and number of particles measured on these filters are used for characterizing the cleanliness of the components, according to international quality requirements.

The following sample preparation steps are used:
(a)Clean the newly manufactured components with oil fluid;(b)Pour the oil through this filter paper so that the particles can be separated out;(c)Dry the filter paper in an oven;(d)Keep the filter paper under a microscope for image acquisition, as shown in Figure 4.


Figure 5a shows the filtration process, after this process it used to get input image captured along with the microscope. Along with steps like threshold, was binarized, and each particle was then further analyzed to obtain the longest dimension of a particle, which was used to determine the particle size; the longer the valid length, the larger the size. Therefore, for this purpose, the binarized version of the image was used to detect the valid length using our software, which automatically finds the longest path in a particle and then measures its length in micron units.

Figure 5b represents processed image while calculating particle sizes and distributions.

Figure 5c represents the stages of detecting the possible lengths of a single particle in the 50 × 50 micron range as the smallest measurable area (box). For each particle, the major axis (length) and minor axis (width) in each grid region are evaluated to obtain the final length of curved particles, where three lines, labeled lengths L1, L2, and L3, are positioned inside the curved shape passing the center of the particle width in each 50 × 50 micron region in order to obtain the final total curved length, L_f_ = L1 + L2 + L3, for the curved particle.

Total Pixel Counting of Particle:

The longest length of the particle is denoted as L. We automatically detected it and then measured its length. This was calculated via the following formula-
Length (Max.) = Total Pixel × Calibration Factor/X Call,
where X Call = actual micron-scale reading.

For finding the particle size, the following apply:

After conducting all the above steps, it becomes easy to find out the particle size of each particle. This is calculated by using the following formula:Particle Size = Total Pixel Count × Calibration Factor.

The same process should be repeated for each detected particle represented with multi colored based on sizes, and the particle sizes of all present particles, as well as their distribution, should also be analyzed and evaluated [21].

## 3. Results

### 3.1. Grain Size Measurement

Stainless steel alloy samples EN8, EN304, EN306, and EN31 were used for experiments. Stainless steel was heat-treated for a 2 h temperature cycle at a temperature of 850 Celsius. After that, sample preparation with stages of mounting, rough polishing, final polishing, and etching with chemical etching methods were carried out. The results were tested using ten samples from each type of specimen. For each sample, the results were obtained from multiple images, with more than 100–150 images tested for each sample using both intercept and planimetric methods. The samples were also tested in NABL-accredited laboratories for accuracy verification, and the results were found to be satisfactory. Figure 6, Figure 7, Figure 8 and Figure 9 show some of the comparison results with average identified grain sizes.

### 3.2. Particle Size Measurement

Stainless steel alloy samples EN8, EN304, EN306, and EN31 were used, and stainless-steel manufactured parts were analyzed for particle size analysis. The results shown in Figure 10, Figure 11 and Figure 12 were observed for the EN 306 sample, and results are classified as the metallic, non-metallic or fiber particles detected, while readings were also verified in NABL-accredited laboratories.

### 3.3. Inclusion Measurements

Inclusion analysis can be conducted to determine the material composition of an image, using ASTM standards [22,23] to estimate the presence of inclusions, porosity, cracks, and flakes. Accurate results can be obtained by separating and categorizing inclusions and analyzing their shape. Along with image processing, artificial intelligence can also be used to detect porosity, inclusions, and scratches on the sample. The size of non-spherical particles, such as needle-shaped crystals or cracks, cannot be accurately measured solely by size. Therefore, methods must be utilized to identify and count agglomerates, too-large particles, and contaminant particles. Porosity, cracks, dendrites, and nodule measurements are found to be super-volatile and depend on many manual factors. Many preprocessing and image enhancement techniques are tested for basic image enhancements. However, these require manual intervention to obtain better results and ensure full autonomy. Figure 13 shows the results of the inclusion measurements of the metal samples. The phase analysis of metal samples is represented in Table 1.

Processed image Figure 13b shows combinations of ferrite, perlite and martensite represented with colors while processing phase analysis. 

### 3.4. Porosity Measurements

The aluminum sample and porosity percentages are shown in Table 2, with one of the processed images in Figure 14.

The porosity percentage (ϕ) of an aluminum sample is calculated by comparing the volume of the pores (Vp) to the total volume of the sample (Vt) using image analysis methods. The evaluation of the porosity characteristics includes determining the percentage of porosity, as well as parameters such as pore density, maximum pore diameter, and pore size distribution.

Processed image Figure 14b shows actual porosity which is highlighted in color while processing porosity percentage measurements. 

### 3.5. Nodularity Measurements

Nodularity refers to the proportion of graphite in spherical (nodular) form compared to other forms like flakes or vermicular (worm-like) graphite. The nodularity percentage is determined by comparing the area occupied by nodular graphite to the total area. within image analysis, which can automatically distinguish between different types and calculate the nodularity percentage. It can further classify graphite forms based on parameters like aspect ratio, roundness, and perimeter-to-area ratio. Furthermore, it helps to assess hollowness in cast irons, thereby evaluating the quality of the cast iron material.

Cast iron nodularity percentage and size measurements based on sizes are shown in Figure 15 along with Table 3.

Processed image Figure 15b shows Nodularity count and its type wise distributions with multiple colors while processing Nodularity percentage and types analysis. 

The nodule count per 0.34 mm^2^ = 98 (Type1 + Type2 + Type3).

The total nodule count per 0.34 mm^2^ = 114.

## 4. Discussion and Comparison

The algorithm automatically detects and measure grain boundaries in captured images and provides statistical data such as average grain number. The results obtained with proposed system results were evaluated on more than 200 images and different types of steel alloys; these results were validated with NABL-accredited and -certified laboratory results, as presented in Table 4, Table 5 and Table 6. It is observed that the proposed algorithm gives an acceptable range of results.

Few important factors need to be considered when using computer vision-based sensing technology, including image quality, calibration issues, and the quality of metal alloy being inspected. The quality of the images used in computer vision algorithms can significantly affect the accuracy of grain size detection. Factors such as lighting conditions, image resolution, and contrast can impact the clarity and definition of the grain boundaries, leading to errors in detection.

However, microscopic imaging has several limitations when it comes to measuring particle sizes and obtaining accurate depth information, particularly concerning porosity and nodularity. Real 3D imaging techniques provide better accuracy and depth information compared to traditional 2D methods. Hence, image analysis software that supports 3D reconstruction and quantitative analysis is essential for achieving reliable and accurate measurements.

Further research with diverse combinations of predictions could improve results. Consequently, future work will aim to explore models to minimize the number of calculations needed. There is scope to examine various phases, twins, and other defects in metal.

## 5. Conclusions

The measurement of grain and particle size in metals plays a vital role in various industries, such as manufacturing, materials science, and engineering. By accurately characterizing these dimensions, one can gain insights into the structural properties, which can allow us to predict the overall performance of metals in specified crucial applications. There are several challenges involved in this computer vision-based sensing process, including those in acquiring microscopic images with magnification and high resolution, implementing effective filtering algorithms, addressing edge rounding correction, and the requirement of intelligent algorithms for grain and particle size analysis. The present research utilizes new image processing techniques to measure grain size. It includes pre-processing, and grain boundary and grain size extraction using an image reconstruction algorithm and fuzzy logic. Noise due to any defects present in an image was also removed during metallographic preprocessing. Edge rounding correction is necessary when working with higher image resolutions as it helps to prevent the issue of obtaining a defocused image with undefined edges. This is because at higher resolutions, i.e., at 100×, the edges of an image can become more pronounced and defined, making it easier to notice any blurriness or a lack of sharpness. By implementing edge rounding correction, the edges are smoothed and refined, resulting in a more polished and professional-looking final product that is free from defocused edges.

In conclusion, the development of a new algorithm for accurately determining the grain and particle size of metals represents a significant advancement in materials science and metallurgy. The results are verified with results obtained from an NABL-accredited testing laboratory. The algorithm’s ability to provide more accurate and reliable results is crucial for various industries, including manufacturing and materials engineering, where precise knowledge of grain and particle size is essential for optimizing the performance and properties of metallic materials. Future research should focus on conducting extensive field trials and case studies with the integration of computational modeling and machine learning algorithms for sensing particle size and properties under different operating conditions and in different metal compositions [24,25,26,27,28,29,30,31,32,33,34,35].

## Figures and Tables

**Figure 1 sensors-24-05328-f001:**
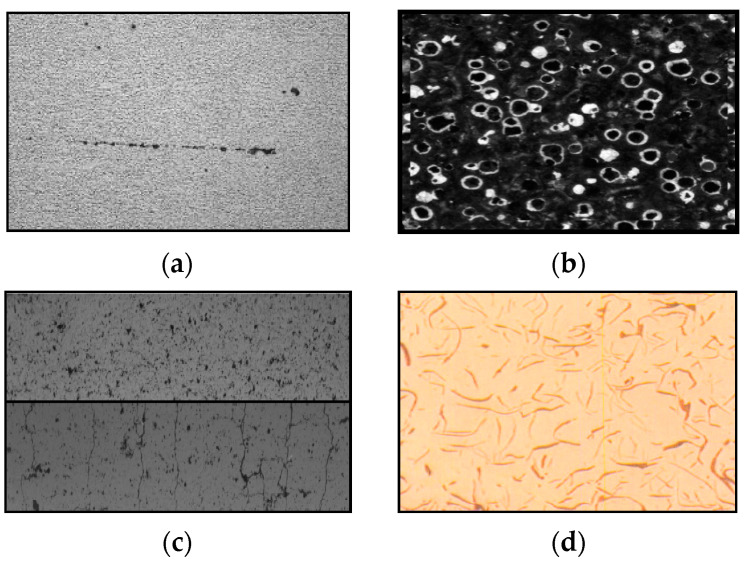
Various quality parameters in metallic images: (**a**) inclusion in metal, (**b**) porosity in metal, (**c**) cracks in metal, (**d**) flakes in metal.

**Figure 2 sensors-24-05328-f002:**
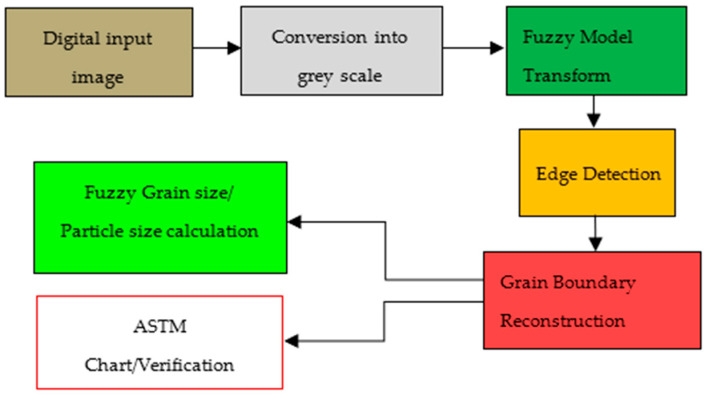
Block diagram of the research work for finding grain size and particle size.

**Figure 3 sensors-24-05328-f003:**
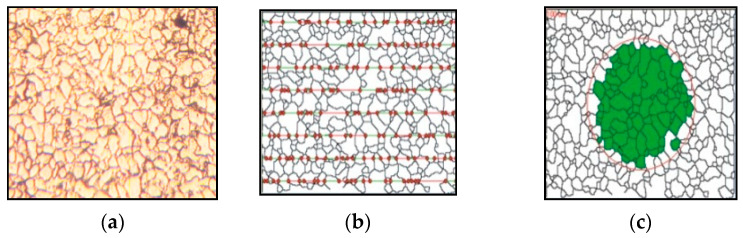
(**a**) Original image, (**b**) intercept method, and (**c**) planimetric method [11].

**Figure 4 sensors-24-05328-f004:**
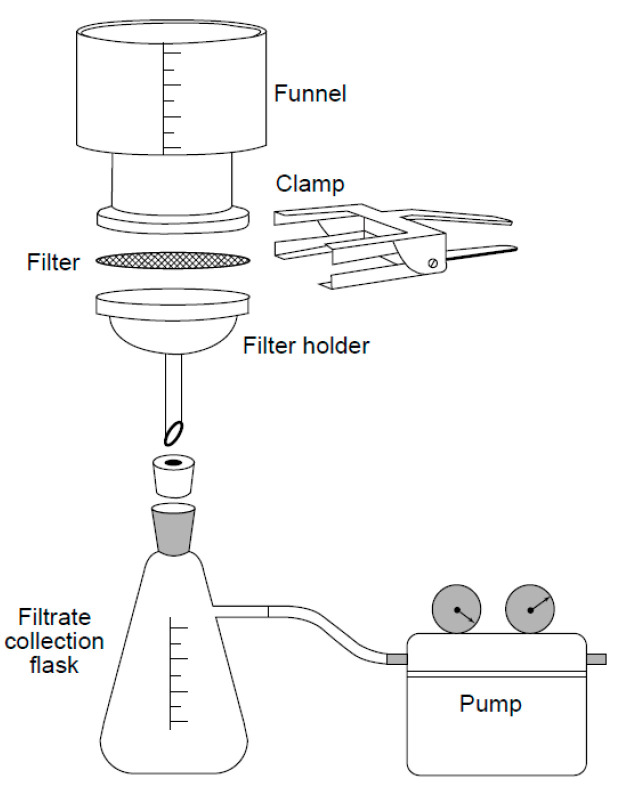
Filtration method [21].

**Figure 5 sensors-24-05328-f005:**
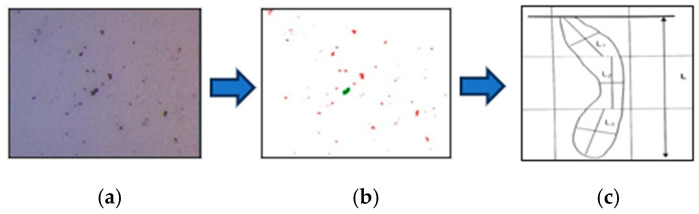
(**a**) Original particle size (**b**), processed particle size, and (**c**) measurement of longest dimension of particle.

**Figure 6 sensors-24-05328-f006:**
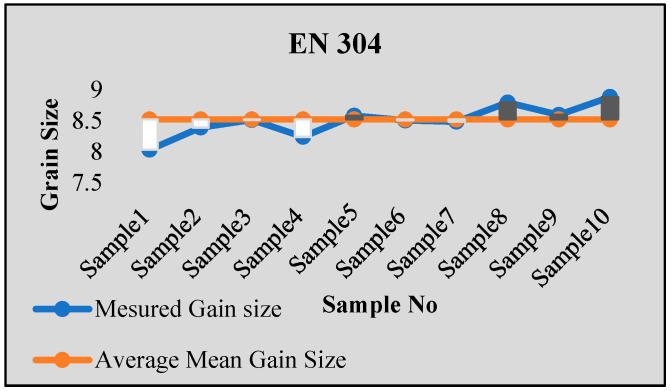
Results for EN 304.

**Figure 7 sensors-24-05328-f007:**
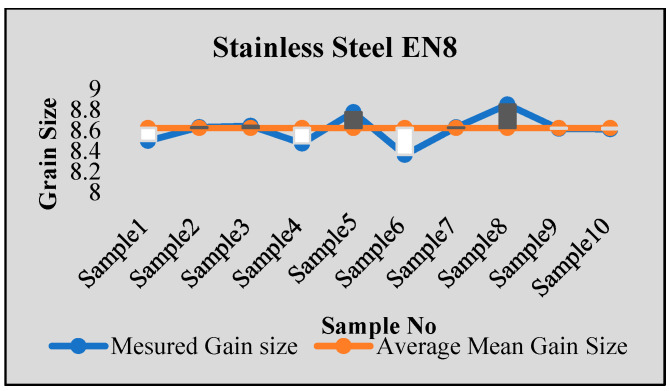
Results for steel EN8.

**Figure 8 sensors-24-05328-f008:**
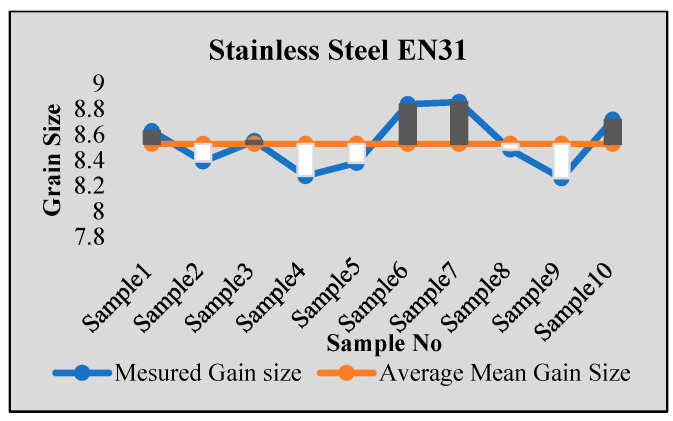
Results for steel EN31.

**Figure 9 sensors-24-05328-f009:**
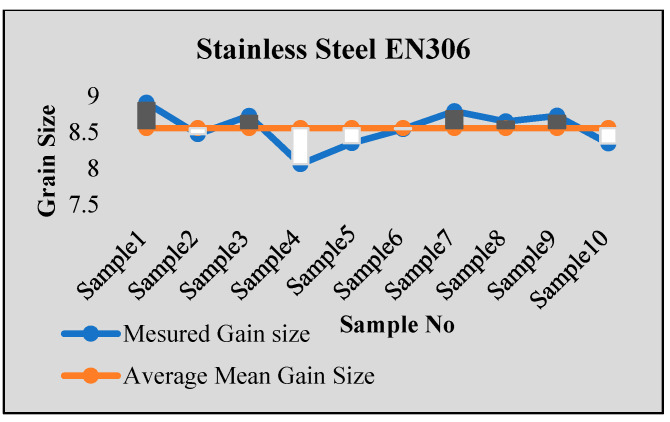
Results for steel EN306.

**Figure 10 sensors-24-05328-f010:**
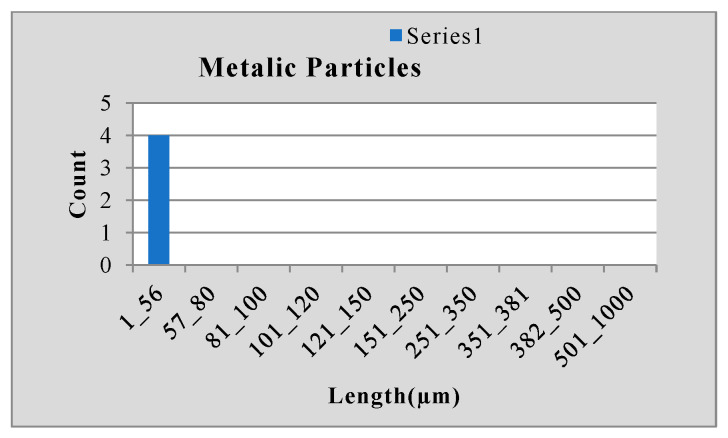
Results for metallic particle size.

**Figure 11 sensors-24-05328-f011:**
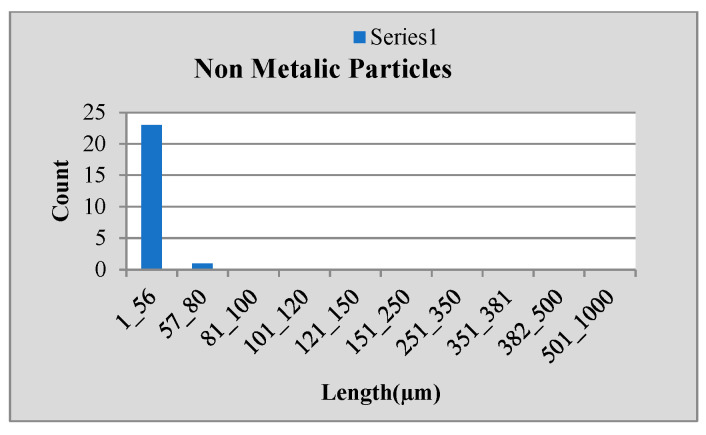
Results for non-metallic particle size.

**Figure 12 sensors-24-05328-f012:**
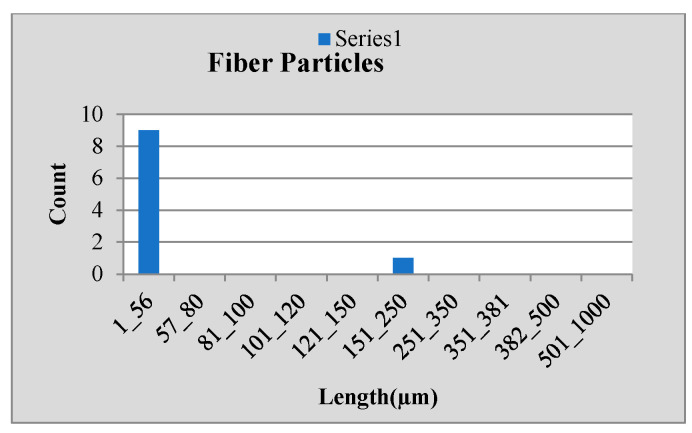
Results for fiber particle size.

**Figure 13 sensors-24-05328-f013:**
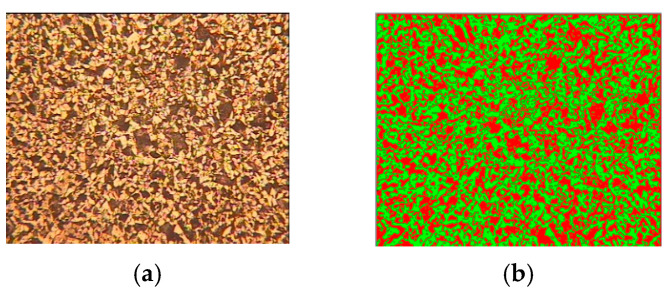
Inclusion measurements for metal samples: (**a**) original image 1; (**b**) processed image 1.

**Figure 14 sensors-24-05328-f014:**
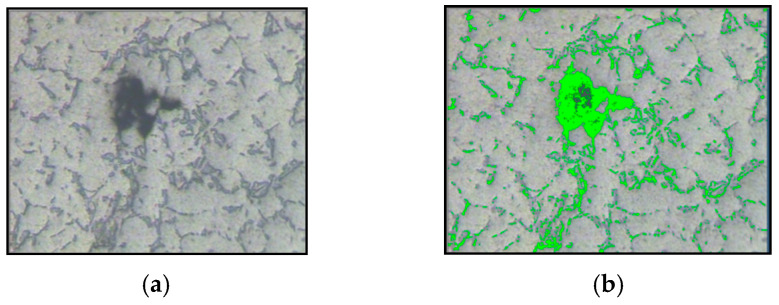
Porosity percentage measurement: (**a**) original porosity image; (**b**) processed porosity image.

**Figure 15 sensors-24-05328-f015:**
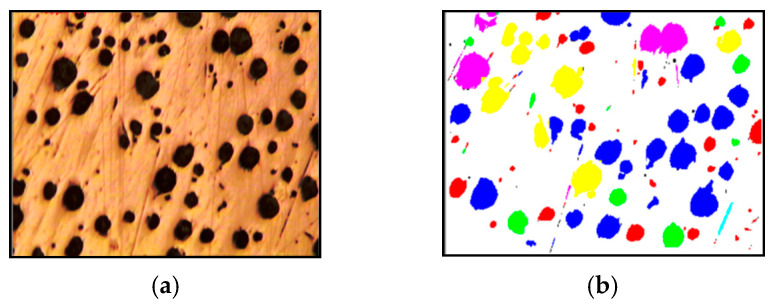
Nodularity percentage measurement: (**a**) original nodularity image; (**b**) processed nodularity image.

**Table 1 sensors-24-05328-t001:** Results of phase analysis.

Sr. No	Metal Alloy Sample Images	Phase 1% Ferrite	Phase 2% Pearlite	Phase 3% Martensite
1	Image 1	46.12	53.84	0.26
2	Image 2	46.01	53.9	0.09
3	Image 3	45.89	53.75	0.36
4	Image 4	45.69	53.65	0.26
5	Image 5	47.04	53.25	0.39
6	Image 6	45.99	54.65	0.24
	Average	46.123	53.84	0.266

**Table 2 sensors-24-05328-t002:** Results of porosity analysis.

Sr. No	Aluminum Sample Images	Porosity%
1	Sample Image 1	8.31
2	Sample Image 2	6.81
3	Sample Image 3	6.83
4	Sample Image 4	9.21
5	Sample Image 5	10.31

**Table 3 sensors-24-05328-t003:** Results of nodularity analysis.

Sr. No	Nodularity Type	Nodularity Count
1	Type 1	58
2	Type 2	13
3	Type 3	27
4	Type 4	9
5	Type 5	6
6	Type 6	1

**Table 4 sensors-24-05328-t004:** Grain size comparison.

Sr. No	Sample Name	Common Name	Obtained Grain Size with Proposed Method	Grain Size Certified by NABL Lab
1	EN8	Hot-rolled carbon steel	8.6	7–8
2	EN304	Austenitic stainless steel	8.5	7–8
3	EN306	Austenitic stainless steel	8.5	7–8
4	EN31	Austenitic stainless steel (different grades)	8.5	7–8

**Table 5 sensors-24-05328-t005:** Particle size comparison.

Sr. No	Sample Name	Common Name	Obtained Particle Size with Proposed Method	Grain Size Certified by NABL Lab
1	EN306	Austenitic stainless steel	56 µm	45–125 microns

**Table 6 sensors-24-05328-t006:** Quality assurance parameter comparison.

Sr. No	Measurements with Metal Alloy EN306		Obtained Results with Proposed Method	Standard Results from NABL Lab
1	Inclusions	Phase 1% Ferrite	46.123	30–50
		Phase 2% Pearlite	53.84	10–25
		Phase 3% Martensite	0.266	0–10
2	Porosity Percentage		8.29	8–9
3	Nodularity count		1–58	1–55

## Data Availability

The data presented in this study are available on request from the corresponding author.

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
