# Peer review of "Revolutionizing Grain and Particle Size Measurement in Metals: The Role of Sensor-Assisted Metallographic Image Analysis"

_sensors, 2024, doi:10.3390/s24165328_

Round 1

Reviewer 1 Report

Comments and Suggestions for Authors

The references looks too old. It looks like using some traditional method of profile extraction or Edge detection. The author may discuss the difference or effectiveness of normal ways of edge detection, including Sobel, Canny,  Prewitt, Roberts, and fuzzy logic methods. The numbering of the figures looks strange. The format of the formulas is also strange. The authors may have to add some more viewpoints or details in the discussion section, which shows little viewpoints.

Comments on the Quality of English Language

Moderate revising of the English is required.

Author Response

Reviewer 1:

The references look too old.

Response: Additional recent references are now included and compared to increase the quality of the paper.

It looks like using some traditional method of profile extraction or Edge detection. The author may discuss the difference or effectiveness of normal ways of edge detection, including Sobel, Canny,  Prewitt, Roberts, and fuzzy logic methods.

Response: Yes, We have used fuzzy logic method in as explained in 2.1.2.

The numbering of the figures looks strange. The format of the formulas is also strange.

Response: The figure numbering and format is now revised as per MDPI style. Total number of figures is now only 15.

The authors may have to add some more viewpoints or details in the discussion section, which shows little viewpoints

Response: The discussion section is now elaborated.

Reviewer 2 Report

Comments and Suggestions for Authors

The topic of this paper is so interesting but there are problems need to be corrected. What is the "Sensor-Assisted Metallographic Image Analysis"? Is a software or equipment? What kind of sensors which you mention? The quality of metallographic processes are an important factor for examining different cracks, porosities, inclusions, and flakes. Can the "Sensor-Assisted Metallographic Image Analysis" examine various phases, twins, and other defects? The cited references are so few and authors should add amount of references above 30~40. More newer references within 3 years must be added to this paper.  The contents of the Sections 3.2, 3.4, and 4 are too short. 3.4. Nodularity Measurements should be 3.5, not 3.4. There is not enough on the academic depth and authors must increase the academic strength. Therefore, I suggest that this paper will be reconsidered after major revisions according to above all reasons. 

Author Response

Reviewer 2:

The topic of this paper is so interesting but there are problems need to be corrected. What is the "Sensor-Assisted Metallographic Image Analysis"? Is a software or equipment? What kind of sensors which you mention? The quality of metallographic processes are an important factor for examining different cracks, porosities, inclusions, and flakes. Can the "Sensor-Assisted Metallographic Image Analysis" examine various phases, twins, and other defects?

Response: It is a system, which collect the input metal images from ultramicroscope. The image sample is collected after pre-processing on metals to be investigated. The collected image is analysed with the help of developed software systems. The grain and particle size estimations are done by the developed software. Micro-structural analysis is necessary to ensure the quality of the material, involving regular inclusion measurements and critical particle and grain size assessments for particulate samples.

The developed system is capable of analysing the composition of the metals by the way of calculating its Grain size, Inclusions Measurements, Porosity Analysis, Nodularity Analysis. It is possible to with the help of sensor assisted Metallographic Image Analysis to examine various phases, twins, and other defects also. Presently that part is included in future scope of the research work to give directions to follower of this paper.

The cited references are so few and authors should add amount of references above 30~40. More newer references within 3 years must be added to this paper.

Response: The number of references is now increased to 35. We have also included newer references within last 3 years in this paper.

The contents of the Sections 3.2, 3.4, and 4 are too short. 3.4. Nodularity Measurements should be 3.5, not 3.4. There is not enough on the academic depth and authors must increase the academic strength. Therefore, I suggest that this paper will be reconsidered after major revisions according to above all reasons. 

Response: The sections 3.2, 3.4, and 4 are now elaborated to have enough depth of academic understanding to the readers. The section 3.4. Nodularity Measurements  is now corrected to 3.5

Reviewer 3 Report

Comments and Suggestions for Authors

In this paper, A final product's quality, durability, and lifespan are determined by the quality analysis of

properties of the material with the proper tolerances. A flaw in a single component can cause the  

breakdown of the entire finished product. Micro-structural to uphold quality, inclusion measure-  

ment is a routinely required process. The work is interesting and useful for metallurgigraphic readers. After answering the following questions, it can be published in sensors:

 In Consider the given size of Image MxN is 165x124. here calculating signal should be changed rather than x letter.

 Line 255, a) Clan the newly manufactured components with oil fluid. is not easy to be understood. Please explain the clan method in details.

 Page 8, the figure 11 should be drawn in more clear shape. Moreover, the usage of L1, L2, and L3 should be explain clearly.

The text to explain figures 9, 10 and 11 should be rewritten, which seems a little confused.

For a better presentation, some figures can be united/merged into one larger figure to give a clearer order.

The format of references should be unified. 

Comments on the Quality of English Language

ok

Author Response

Reviewer 3:

In this paper, A final product's quality, durability, and lifespan are determined by the quality analysis of properties of the material with the proper tolerances. A flaw in a single component can cause the  breakdown of the entire finished product. Micro-structural to uphold quality, inclusion measurement is a routinely required process. The work is interesting and useful for metallurgigraphic readers. After answering the following questions, it can be published in sensors:

In “Consider the given size of Image MxN is 165x124.” here calculating signal should be changed rather than “x” letter.

Response: Yes, it’s now corrected

Line 255, “a) Clan the newly manufactured components with oil fluid.” is not easy to be understood. Please explain the clan method in details.

Response: Yes, the detail of the process of cleaning and filtration is now added along with figures for more clarity.

Page 8, the figure 11 should be drawn in more clear shape. Moreover, the usage of L1, L2, and L3 should be explain clearly.

Response: Yes, It is now redrawn and necessary explanation of use of L1, L2, and L3 is now added.

The text to explain figures 9, 10 and 11 should be rewritten, which seems a little confused.

Response: Yes, it is now revised.

For a better presentation, some figures can be united/merged into one larger figure to give a clearer order.

Response: Yes, it is now revised

The format of references should be unified.

Response: Yes, it is now unified as per MDPI reference style.

Round 2

Reviewer 2 Report

Comments and Suggestions for Authors

I checked the revised paper which was well written. Therefore, I accepted this paper for publication in Sensors without further revision. 

Reviewer 3 Report

Comments and Suggestions for Authors

my comments are fully addressed.